# In Vitro Heme Coordination of a Dye-Decolorizing Peroxidase—The Interplay of Key Amino Acids, pH, Buffer and Glycerol

**DOI:** 10.3390/ijms22189849

**Published:** 2021-09-12

**Authors:** Kevin Nys, Vera Pfanzagl, Jeroen Roefs, Christian Obinger, Sabine Van Doorslaer

**Affiliations:** 1BIMEF Laboratory, Department of Chemistry, University of Antwerp, 2610 Antwerp, Belgium; kevin.nys@hotmail.com (K.N.); jeroen@roefs.be (J.R.); 2Division of Biochemistry, Department of Chemistry, BOKU—University of Natural Resources and Life Sciences, 1190 Vienna, Austria; vera.pfanzagl@boku.ac.at (V.P.); Christian.obinger@boku.ac.at (C.O.)

**Keywords:** heme peroxidases, electron paramagnetic resonance, active site structure, UV-vis spectroscopy, alkaline transition, ligand binding, glassing agents

## Abstract

Dye-decolorizing peroxidases (DyPs) have gained interest for their ability to oxidize anthraquinone-derived dyes and lignin model compounds. Spectroscopic techniques, such as electron paramagnetic resonance and optical absorption spectroscopy, provide main tools to study how the enzymatic function is linked to the heme-pocket architecture, provided the experimental conditions are carefully chosen. Here, these techniques are used to investigate the effect of active site perturbations on the structure of ferric P-class DyP from *Klebsiella pneumoniae* (K*p*DyP) and three variants of the main distal residues (D143A, R232A and D143A/R232A). Arg-232 is found to be important for maintaining the heme distal architecture and essential to facilitate an alkaline transition. The latter is promoted in absence of Asp-143. Furthermore, the non-innocent effect of the buffer choice and addition of the cryoprotectant glycerol is shown. However, while unavoidable or indiscriminate experimental conditions are pitfalls, careful comparison of the effects of different exogenous molecules on the electronic structure and spin state of the heme iron contains information about the inherent flexibility of the heme pocket. The interplay between structural flexibility, key amino acids, pH, temperature, buffer and glycerol during in vitro spectroscopic studies is discussed with respect to the poor peroxidase activity of bacterial P-class DyPs.

## 1. Introduction

Heme-containing proteins are widespread among all kingdoms of life. Their function is largely determined by the folding architecture of the peptide chain, modulating the possible redox properties of the heme pocket as well as its solvent and substrate accessibility [1,2]. Perhaps not surprisingly, this results in an intriguingly high level of functional versatility. It has become a major goal in heme-protein research to unravel how the enzymatic functions are governed by the local environment [2]. To achieve this, a wide variety of complementary techniques has been developed for structural and electronic characterization [3]. While X-ray crystallography or NMR spectroscopy provide insight into the protein structure, electron paramagnetic resonance (EPR) allows probing (transient) paramagnetic intermediates of the active site, thus exploring key mechanistic steps in the enzymatic cycle or protein function [4,5]. When performing EPR spectroscopy on heme proteins, one should, however, keep in mind that external factors possibly influence the observations of the active site and thus induce illegitimate conclusions. In this context, awareness was raised by Svistunenko et al. in a low-temperature continuous-wave (CW) EPR study on the ferric heme forms of *Mycobacterium tuberculosis* catalase-peroxidase (*Mt*KatG) [6]. It was shown that not only pH, but also buffer type could alter the electronic architecture of the heme cavity. Earlier studies report on freezing-induced distortions of the heme cavity and stabilizing effects of commonly used glassing agents, such as glycerol [7,8,9,10]. Here, we use multi-frequency EPR to understand pH-dependent changes in the heme pocket of *Kp*DyP, a dye-decolorizing peroxidase from the human pathogen *Klebsiella pneumoniae*.

Dye-decolorizing peroxidases or DyPs are hydrogen-peroxide dependent oxidoreductases of predominantly bacterial origin, which are further classified in three distinct classes (P, I and V with *Kp*DyP belonging to the P-class DyPs (formerly B-class)) [11]. Bacterial P-class DyPs are very poor peroxidases with unknown biological function. Hydrogen peroxide efficiently mediates the rapid formation of Compound I, which is remarkably stable and shows only modest reactivity towards organic and inorganic electron donors [12,13]. *Kp*DyP has been biochemically [12] and structurally [13,14] well characterized and serves as a good model for this protein family, which has gained significant interest in recent years in regard to their biotechnological potential [15]. The conserved core fold of DyPs (four beta sheets connected by alpha helices) classifies them within the dimeric α + β-structural superfamily together with the phylogenetically related enzymes chlorite dismutases (Clds) and coproheme decarboxylases (ChdCs) [16,17,18]. All three protein families have a characteristic loop, which forms the outer wall of the active site and dictates accessibility and shape of the cavity [12,13,19,20,21]. In DyPs this loop furthermore contains one of the main catalytic residues of the distal side, Asp-143 in *Kp*DyP, which was shown to be required for heterolytic cleavage of hydrogen peroxide [12]. The equally conserved distal arginine, Arg-232 in *Kp*DyP, is highly important to maintain the integrity of the active site. The crystal structure of wild-type (WT) *Kp*DyP in Figure 1 shows the extensive hydrogen bonding network connecting Arg-232 and Asp-143 with propionate p6 and active site waters. Limited access to the heme iron through the main channel (Figure 1, green) whose bottleneck is formed by the catalytic Asp-Arg pair together with Phe-248 and Leu-246 was proposed to be the main reason for the observed low catalytic activity of this enzyme [12]. This channel is proposed to be the main entry route for hydrogen peroxide required for Compound I formation. Previous studies showed that the overall structure of the active site is maintained in a D143A variant but that this mutation increases the bottleneck radius [12,13]. Exchange of Arg-232 for alanine (R232A); however, has severe structural consequences, leading to a rearrangement of the loop and collapse of the active site. This effectively removes the main heme iron access route as seen in the crystal structure and molecular dynamics simulations. Taking the crystal structure of the D143A/R232A double variant into account, where the WT-like loop conformation is solely due to a coordinating glycerol molecule, it is reasonable to expect an increased structural susceptibility of these mutants to external factors, such as temperature and buffer components [12].

In order to further investigate the role of the conserved Asp-143/Arg-232 pair in the restriction of the substrate accessibility and ligand binding to the heme iron, we used continuous-wave (CW) and pulsed EPR to explore the active site of WT *Kp*DyP and D143A, R232A and D143A/R232A variants in different buffers at neutral and high alkaline pH, as ligand binding is often enhanced in the alkaline region due to deprotonation. The buffers were chosen based on their widespread use and potential to influence the heme active site. We purposefully omitted the acidic pH range as no specific structural changes are expected other than pH induced unfolding starting from pH 5 [12]. We show the importance of the conserved distal arginine for accommodating a hydroxo ligand at alkaline pH. Furthermore, our results challenge the non-innocent role of glycerol and buffer molecules on heme proteins with an accessible active site. While the glassing agent glycerol is commonly added to prevent freezing artefacts at the low temperatures required for the EPR experiment [23], it inhibits certain ligation states in the *Kp*DyP variants under study and with related spectral changes in the EPR and optical absorption spectra. Moreover, we highlight a strong effect of the type of buffer on the *Kp*DyP variants. An alkaline transition is not consistently maintained for different buffers and use of a glycine-containing buffer at high pH induced ligation of the ferric heme iron with glycine. Although slight effects of buffer molecules on the EPR spectra of heme proteins have been reported before [6,24], this is the first time that glycine binding to heme iron is clearly identified at low temperatures (EPR) and room temperature (optical absorption). We discuss how the differing ligand accessibility in WT KpDyP and its variants links to the proposed reaction mechanism of the wild-type protein.

## 2. Results

### 2.1. Optical Absorption Spectroscopy

Figure 2 (solid lines) depicts the optical absorption spectra of WT *Kp*DyP and the D143A, R232A and D143A/R232A variants in phosphate buffer (pH 7.0), borate buffer (pH 10.0) and glycine-KOH buffer (pH 10.0) taken at room temperature. The phosphate buffer was chosen in accordance with previous studies on WT KpDyP [12]. As reported earlier [12], at pH 7 WT *Kp*DyP exhibits a Soret peak at 406 nm with a broad shoulder at the lower wavelength side, Q-bands at 508 and 540 nm and a charge transfer (CT) band at 641 nm, in accordance with a high-spin (HS) Fe(III) state of the heme iron. The same spectrum is found for the D143A variant, with a narrower Soret peak at 406 nm and slightly blue-shifted CT band (630 nm) for the R232A and D143A/R232A variants. While the latter spectral features agree with an *S* = 5/2 Fe(III) state, the absorption spectra of WT KpDyP and D143A may indicate some degree of quantum mixing of the *S* = 5/2 state with an intermediate *S* = 3/2 state [25]. Addition of 25% glycerol (Figure 2, dashed lines) induces a narrowing of the Soret peak with concomitant shift of the CT peak for the D143A variant, while it does not affect the absorption spectra of the other variants (Figure 2). 

In borate buffer (pH 10) without glycerol, WT, R232A and D143A/R232A *Kp*DyP exhibit the same optical absorption spectra as at pH 7, but a marked change is observed for the D143A variant indicating the presence of HS and low-spin (LS, *S* = 1/2) ferric heme states. LS ferric states occur when a strong base, such as a hydroxo-anion or imidazole, is ligating the heme iron. Previous work has shown that an alkaline transition to a hydroxo-ligated heme species takes place for this variant, with the absorption spectrum of this species exhibiting a Soret peak at 410 nm and additional bands at 540, 576 and 610 nm [12]. The presence of this LS species is inhibited by the addition of glycerol. 

The non-innocent effect of the buffer is noticed when a glycine-KOH buffer is used (Figure 2). While WT and R232A *Kp*DyP maintain the HS-state, a significant change in the optical absorption spectrum is noticed for the variants where Asp-143 is exchanged for an alanine. Here, a new LS state is observed with D143A showing a red-shifted Soret peak at 412 nm and Q-bands at 532 and 561 nm (double variant: 413, 532 and 562 nm). The CT peak is no longer visible demonstrating the absence of a HS state. Addition of glycerol does not alter the UV-visible spectral signatures in this case.

### 2.2. Influence of Buffer and Glycerol on CW EPR Spectra

The optical absorption analysis reveals a worrying non-innocent effect of glycerol and the type of buffer molecules on the heme ligation in the *Kp*DyP variants under study. To understand this further and to relate this to the protein’s heme-pocket architecture and behavior, we performed X-band continuous-wave (CW) EPR. Moreover, glycerol is commonly used in EPR as a cryoprotectant and is in some cases essential to avoid strong dipolar interactions between the paramagnetic centers [23]. 

Figure 3A shows the X-band CW EPR spectra of WT *Kp*DyP and variants in phosphate buffer (pH 7.0) without glycerol. Figure 3B shows the comparative results with glycerol in accordance with our earlier reported data [12]. It is clear that the site-directed mutations as well as the cryoprotectant cause changes to the electronic structure of the active site that go beyond the minor modifications that are expected because of the change in dielectric constant. As can be derived from the EPR parameters given in Appendix A, differences include a variation in abundancy of specific species, a change in rhombicity of the zero-field tensor (*E*/*D*) and the appearance of extra HS species. The double variant exhibits an even more striking change, since it reveals formation of an LS species (~24%) that disappears with the addition of glycerol (see inset Figure 3A, LS1 in Appendix A). The origin of this contribution is unknown so far and cannot be deduced solely from the *g*-values. However, this LS species was not observed in the optical absorption spectra (Figure 2) and is thus due to a ligation induced by the low temperatures. The appearance of HS species with *E*/*D* values ≥ 0.017 seems to agree with the observation of a broader Soret band and higher wavelength and broadening of the CT band in the optical absorption spectra (Figure 2). This correlation is not unexpected, since the increase of the *E*/*D* parameter and the concomitant decrease of $g_{12}=\left(g_{x}^{eff}+g_{y}^{eff}\right)/2$ has been linked to an increased admixture of the *S* = 5/2 and *S* = 3/2 states. According to the concepts outlined by Maltempo [26], the *g*_12_ values associated with the largest *E*/*D* values found here (*g*_12_ = 5.83 to 5.685) can be related to 10–15% of the *S* = 3/2 state.

Furthermore, without addition of glycerol, the spectral shape of the EPR signal of the HS components in WT *Kp*DyP is found to be strongly batch-dependent (Figure 4A). Only after addition of the glassing agent, all batches exhibit the same spectral signature (Figure 4B). 

Note that the stable organic radical, observed earlier in the resting state of WT *Kp*DyP (Figure 3, asterisk [12]), is not affected by the addition of glycerol.

At alkaline pH, several heme proteins are known to display a distal coordination of a hydroxo ligand [27,28,29]. For *Kp*DyP variants, it appears this is only the case under specific conditions as was illustrated by the optical absorption spectroscopy (Figure 2). The CW EPR spectra of *Kp*DyP and variants in borate buffer at pH 10.0 (Figure 5) mostly agree with the optical absorption data. In general, the HS Fe(III) state is maintained in all frozen solutions both with and without glycerol, although the specific type and weight of the HS contributions again depends on whether glycerol is added or not (Figure 5, Appendix A. Moreover, the change of buffer and pH influences the electronic structure of the paramagnetic center. A careful inspection of the EPR parameters obtained from the spectra in Figure 5 (Appendix A) and Figure 3 (Appendix A) reveals that a more alkaline pH provokes a decrease in abundancy of the HS components with higher rhombicity (*E/D* ≥ 0.017). This is in line with optical absorption spectra, where a broadened Soret band is only found for WT *Kp*DyP (Figure 2).

Appendix A shows an enlargement of the EPR spectra of Figure 5 in the (180–400)mT range, revealing the additional presence of contributions of different LS species. For both WT *Kp*DyP and the D143A variant (without glycerol, Appendix A) we find contributions of hydroxo-ligated ferric heme species. In the former this only accounts for 4% of all the ferric heme contributions, but the EPR spectrum of D143A displays three different contributions of hydroxo-ligated heme species that together account for ~30% (Appendix A). This agrees with the contribution of a hydroxo-ligated ferric form in the optical absorption spectrum of D143A (Figure 2). The *g*-values of the different hydroxo-ligated heme species are summarized in Table 1. While the *g*-tensor of species OH1′ is very similar to the one found for horseradish peroxidase (HRP) [30], the parameters of OH2/OH2′ and OH3′ are more common for alkaline forms of cytochrome *c* peroxidase and mammalian myoglobin [24,29,31]. A high diversity of hydroxo-ligated species was found in hemoglobin of *Thermobifida fusca* [32] and the globin domain of the GLB-33 globin of *C. elegans* [33]. The Arg-232 variants (both single and double variant) do not exhibit hydroxo-ligated species. However, all but the D143A variant show a contribution (labeled LS2) with *g*-values ascribed to binding of a glutathione or a buffer molecule in other heme proteins [34,35]. In addition, the EPR spectrum of the double variant reveals the same LS form of unknown nature (LS1) observed at pH 7. Addition of glycerol removes all LS contributions, except for the LS2 form in the R232A variant. Addition of 30% glycerol reduces the pH of borate buffer to pH 7 [36], where all variants are predominantly HS. The LS2 form, retained in R232A, is thus likely a buffer molecule, as it is not observed in phosphate buffer.

The optical absorption spectra (Figure 2) showed a drastic effect of the use of a glycine-KOH buffer at pH 10. Both D134A and the double variant show maxima at 533 nm and 574 nm, which are distinct from the absorption maxima typical for OH^−^ ligation observed for D134A in borate buffer (540, 574 and 610 nm). This is confirmed in the corresponding EPR spectra (Figure 6). Without addition of glycerol, only the EPR spectrum of the WT enzyme shows an appreciable contribution of HS ferric heme forms (~27%) (Figure 6A). The majority of the EPR signal of WT *Kp*DyP stems from a LS Fe(III) species with maximum *g*-value 3.27 (species Gly2 in Appendix A). Similar contributions are also present in the R232A and D143A variants (Figure 6A, Appendix A). In addition, LS species with *g_z_* ≈ 3.11–3.14 and *g_y_* ≈ 2.09–2.11 are observed in all single variants, which we indicate as Gly1-type signals (Appendix A). For the double variant, the contribution of a single LS species with *g*-values in between those of Gly1 and Gly2 is observed (Gly2′ in Appendix A). Appendix A shows a detailed comparison of the LS species.

Upon addition of glycerol, the Gly2 contribution disappears fully from the EPR spectrum of WT *Kp*DyP and the HS ferric heme forms again dominate the spectrum (Figure 6B and Appendix A). Similarly, the EPR spectrum of R232A *Kp*DyP is governed by HS forms, while only the Gly2 LS contribution remains present in the EPR spectrum (Appendix A). The D143A variant exhibits a less appreciable influence of glycerol addition with both LS features only shifting in relative amount and no appearance of a HS contribution (Appendix A). Finally, the EPR spectrum of the double variant remains essentially the same upon addition of the cryoprotectant, with small shifts in the *g*-values that can be attributed to the change in the dielectric constant. The *g*-values point to a Gly2-type, rather than Gly1-type contribution (Appendix A). Note that the optical absorption spectra revealed a dominant HS form for WT and R232A *Kp*DyP in this buffer, while for the variants in which the Asp-143 is mutated only the LS heme form is detected independently of the presence of glycerol (Figure 2). This highlights that the glycerol-dependent LS↔HS conversion observed by EPR for WT and R232A *Kp*DyP is to some extent also induced by the low temperatures. 

W-band electron-spin-echo (ESE) detected EPR experiments were performed in an attempt to facilitate the interpretation of the X-band CW EPR shown in Figure 6B (see Appendix A). Although the experiments essentially confirmed the above analysis of the X-band EPR data, only minor additional information could unfortunately be obtained due to the low spin-echo intensity and the presence of contaminants.

### 2.3. Subtle Changes in the Heme Pocket Revealed by Pulsed EPR

While optical spectroscopy revealed that the heme of each described variant is in a HS Fe(III) state at pH 7 (Figure 2), CW EPR spectroscopy revealed clear differences in the *g* and zero-field parameters of these HS forms (Appendix A, Figure 3), which can be related to subtle changes in the electronic structure of the heme pocket. Mutation of the distal Asp-143 and/or Arg-232 in *Kp*DyP formation of HS species with lower rhombicity of the zero-field parameters (smaller *E/D*) is more favored, especially in the presence of glycerol (Appendix A). 

Here, we will deploy X-band hyperfine sublevel correlation (HYSCORE) spectroscopy to further evaluate the implications of the distal changes on the electronic structure of the cofactor. This 2D pulsed EPR experiment enables the determination of hyperfine and nuclear quadrupole values of the neighboring magnetic nuclei. The latter interaction is present for nuclei with nuclear spin *I* > ½ such as ^14^N.

In HS ferric forms of globins, such as mammalian myoglobin, low *E/D*-values are observed when a water molecule is ligating the heme iron on the distal side [4]. In order to probe whether water ligation induces the observed reduction in the rhombicity of the zero-field values in the *Kp*DyP variants, ^1^H HYSCORE is measured of WT, D143A and R232A *Kp*DyP in glycerol-containing frozen solutions at pH 7 (Figure 7, bottom spectra). If distal water ligation occurs, characteristic cross peaks are found in the ^1^H HYSCORE spectra measured at the high-field magnetic-field position (*g_z_^ef^*^f^) [37,38]. The region in which these cross peaks are expected is indicated in the spectra in Figure 7. It is clear that the HYSCORE spectra of all three *Kp*DyP variants lack this feature and that distal water ligation therefore does not occur in the three proteins under these conditions. 

Figure 7 (top panel) shows the (−, +) quadrant of the HYSCORE spectra for WT, D143A and R232A *Kp*DyP variants taken at a magnetic-field setting corresponding to *g_z_^eff^*. The observed cross peaks correspond to the interaction between the electron spin and the ^14^N heme nuclei, in accordance with observations for other HS heme proteins, such as globins [37,39] and peroxidases [38] (Table 2). The interpretation of the individual cross peaks in the HYSCORE spectra is given in the Appendix A). Because of the orientation selection at this magnetic-field setting, the spectral simulation of the cross peaks only reveals the *z*-component (i.e., component along the heme normal) of the hyperfine and quadrupole tensor of the porphyrin ^14^N nuclei (Table 2). While ferric WT and D143A *Kp*DyP show identical ^14^N hyperfine values, a marked change is observed for the R232A variant.

A previous in-depth study of aquometmyoglobin showed that the cross peaks related to the interaction of the electron spin with the N_ε_ of the proximal histidine are usually weak or suppressed in the standard HYSCORE spectra, as is also observed here. At other magnetic-field settings, the matched ^1^H HYSCORE spectrum reveals the characteristic cross-peaks due to the protons at the C_ε_ and C_δ_ positions of the proximal His (Appendix A) [40].

The (0–7, 0–7) MHz region of the HYSCORE spectra at *g = g_z_^eff^* reveals signals from the interaction of the electron spin with weakly coupled ^14^N and ^13^C nuclei, the latter in natural abundance (Appendix A) [37]. Again, marked shifts are found in these couplings upon mutation of the distal Arg-232, while this is not the case for the mutation of the distal Asp-143 (see Appendix A for more details). 

## 3. Discussion

### 3.1. Heme Cavity Heterogeneity and the Influence of Glycerol

At neutral pH, the heme in the homodimeric dye-decolorizing peroxidase of *K. pneumoniae* exhibits a high-spin Fe(III) state both at room temperature and in frozen solution. Mutation of the catalytically important residues Asp-143 and Arg-232 to alanine or the addition of a glassing agent, i.e., glycerol, preserves the oxidation and high-spin state. While this observation may be derived from both optical absorption (Figure 2) and low-temperature EPR spectroscopy (Figure 3), significant differences are observable in the electronic architecture. WT *Kp*DyP and variants display a heterogeneous heme pocket depicted by the presence of contributions of different HS states in the EPR spectra, characterized by a changing rhombicity of the zero-field splitting parameter (Figure 3). An increase in this rhombicity (larger *E/D*-value), which translates in a splitting of the EPR feature at *g ~* 6, reflects a departure from the tetragonal symmetry that is distinctive for the porphyrin and that has been related to an increased admixture with the intermediate *S* = 3/2 state [25,26]. This lowering of the symmetry is governed by the physical environment, i.e., the ligands of the heme and its incorporation in the protein matrix. The HS metal ion is thus able to sense conformational changes of the protein moiety that impact the geometry of the heme cavity [42,43].

For many HS ferric heme proteins, small *E/D*-values are related to the binding of a water molecule at the distal site, as is known to be the case for aquometmyoglobin [3,39]. However, while the EPR spectra of both the Asp-143 and Arg-232 variants of *Kp*DyP contain species with small *E/D*-values (HS3, Appendix A), water binding is contradicted by the HYSCORE data (Figure 7). Our earlier study revealed that WT *Kp*DyP and variants exhibit a pronounced hydrogen-bonding network [12]. The reported presence of multiple water molecules in the heme pocket [12], stabilized by the catalytically important residues and a heme propionate (Figure 1), suggest that different arrangements in the distal heme area are possible and may explain the heterogeneity in the heme cavity observed by EPR. Earlier EPR studies already pointed out that heme peroxidases commonly display a certain variability in their active site [6,10,42,44,45]. Moreover, measurements of in situ horseradish peroxidase (HRP) showed that the HS Fe(III) signals changed their spectral shape upon transfer from a test tube to an EPR tube [42]. And studies on yeast cytochrome *c* peroxidase indicated that an identical preparation of the enzyme gave rise to EPR spectra with a different ratio in LS and HS species [46]. A similar observation was made here with different batches of WT *Kp*DyP (Figure 4A). Cao et al. showed that the freezing and thawing rate in potassium phospate buffers influenced the denaturarion of different proteins [47]. Both surface denaturation and pH shift due to precipitation of the buffer salt are indicated as likely reasons of protein damage during freezing. Note, however, that the observation of high fractions of HS heme species with *E/D* ≥ 0.017 by EPR at low temperature for ferric WT and D143A *Kp*DyP (Appendix A) correlate with the observation of a broadened Soret peak and red shift of the CT band at room temperature optical absorption spectra (Figure 2), indicating that the observed EPR-spectral differences between the variants is not merely an effect of the low temperature.

The variability in the electronic structure of the heme pocket observed by low-temperature EPR can be prevented by the addition of glycerol (Figure 4B). It has been reported before that the use of a glassing agent not only affects the glassing temperature of the solvent but also limits the co-existence of different conformational sub-states of the protein [48]. Resonance Raman spectroscopy of ferric cytochrome *c* peroxidase revealed that glycerol induces a stabilization of the protein structure, accompanied by a reduced interaction of the iron with its distal ligand [49]. Furthermore, glycerol-induced changes have been reported in the EPR spectra of ferric CcmE heme chaperone [41]. In the crystal structure of ferrous as well as ferric *Kp*DyP [12,14] a glycerol molecule was found to be coordinated in the upper region (i.e., above the bottle neck) of the main access channel. Additionally, although *Kp*DyP displays a penta-coordination both with and without glycerol, it is highly likely that the glycerol molecule is coordinated in the active site channel and thus influences the active site hydrogen bonding network, its dynamics and variability in solution. Indeed, addition of glycerol forces the different heterogeneous HS Fe(III) states in different batches into an identical set of sub-states (Figure 4B). Moreover, the crystal structure of the D143A/R232A variant reveals the presence of a glycerol molecule in the heme pocket [12]. In fact, crystals of this double variant could only be obtained with glycerol as part of the crystallization solution and glycerol was believed to be essential to stabilize the heme region in a WT-like manner in place of Arg-232 [12].

### 3.2. The Non-Innocent Effect of Buffer Molecules

The comparison of the optical absorption and EPR spectra of WT *Kp*DyP and variants at pH 10 in two different buffers reveals the non-innocent effect of buffer molecules on the electronic spin state of the heme iron. In a glycine-KOH buffer, the optical absorption spectra show a full conversion to the low-spin state for D143A and D143A/R232A *Kp*DyP but not for WT *Kp*DyP, while this is not observed in a borate buffer at the same pH (Figure 2). EPR reveals that this is due to ligation of a strong base to the heme iron (Figure 6A, Appendix A). The large *g_z_* values (≥3) of species Gly1 and Gly2 (Appendix A) are typical of a highly anisotropic LS heme species [50], and are in line with parameters observed for endogenous and exogenous amine ligands [4]. The difference in EPR parameters between Gly1, Gly2 and Gly2′ probably relate to a different orientation and stabilization of the glycine molecule in the heme cavity. Although addition of glycerol slightly changes the EPR parameters, in line with the earlier discussed ability of glycerol to enter the heme access channel or heme pocket, glycine remains bound to the heme iron in the variants involving exchange of Asp-143 for an alanine (Figure 6B, Appendix A). At low temperature, dominant ligation of glycine to the heme is also found for R232A *Kp*DyP and to a minor extent for WT *Kp*DyP. This ligation is, however, far less resistant to glycerol addition (Figure 6B, Appendix A). Earlier studies of other heme proteins already hinted that the buffer can affect the electronic structure of the active site and even its catalytic activity [6,49]. Interestingly the observed glycine ligation and the effect of glycerol addition seems to be unique to Dye-decolorizing peroxidases, as it was not observed in an in-depth study on myoglobin [36]. Moreover, the glycine ligation appears to be stronger in D143A/R232A than in the R232A. As this cannot be solely due to accessibility of the active site, it suggests that Asp-143 selectively blocks substrate binding. 

### 3.3. Alkaline Transition in KpDyP Variants

From the above, it is clear that a well-considered choice of buffer is key to studying alkaline transitions in dye-decolorizing peroxidases. Alkaline pH does not necessarily imply ligation of a hydroxo ligand to the heme iron, especially if there is no aquomet state present in neutral pH conditions. In borate buffer, an alkaline transition is observed for D143A *Kp*DyP, both at room temperature (Figure 2) and at 10 K (Figure 5A). Moreover, WT *Kp*DyP exhibits a small fraction of hydroxo-ligated species in frozen solution (Figure 5A, Appendix A). If R232 is missing, no hydroxo-ligation can be observed at all, suggesting that R232 is required to stabilize this ligand. It is possible that in the case of WT *Kp*DyP the hydroxo-ligated fraction is either too low at room temperature to be detected in the optical spectrum or that the OH^−^ ligation observed at low temperature originates from packing forces on the heme induced by freezing as is the case for the alkaline state at neutral pH in HRP isozyme A2 [28,30]. Moreover, the pH of buffer solutions is known to be slightly changed upon freezing with the effect being buffer-dependent [51]. Interestingly, no hydroxo ligation is displayed in the variants of *Kp*DyP where Arg-232 is replaced by an alanine. This agrees with the general observation that binding of OH^−^ in heme peroxidases is accommodated by a distal arginine forming a hydrogen bond [52,53,54]. The presence of hydrogen bonding can be deduced from the *g_z_* component that reflects the strength of the stabilization [33]. The most dominant contribution in D143A *Kp*DyP, OH1′, shows *g* values very similar to the values found for HRP isozyme A2 and lignin peroxidase isozyme H2 [27,28,30] and is indicative of strong hydrogen bonds between the hydroxo ligand and the distal Arg-232. On the other hand, there is a small contribution of OH3′ with a significantly lower *g_max_* in the range that is common for globins [29]. A third component is observed in both the WT and D143A variant and appears to be a species where the hydroxo ligand experiences a stabilization with intermediate strength. 

Both optical absorption and EPR spectroscopy reveal no alkaline transitions in the *Kp*DyP variants in the presence of glycerol. It is reported, however, that the acidity of a boric acid solution is lowered upon addition of glycerol, possibly explaining the inhibition of an alkaline transition [55,56]. 

Comparison of Appendix A shows that the nature of the HS heme species is also affected by the pH. In general, when the HS Fe(III) state is maintained in borate buffer, the rhombicity of the co-existing species changes (see Appendix A). A study on the ferric heme forms of the catalase peroxidase (KatG) of *Mycobacterium tuberculosis* reported an increase in rhombicity with increasing pH [6]. For *Kp*DyP, we generally observe the opposite behavior, showing that these trends are not uniform across various peroxidase structural families. 

### 3.4. The Distal Heme Side in KpDyP

In the previous section, we highlighted the importance of Arg-232 for stabilizing a distal hydroxo ligand at high pH, as well as its general importance in maintaining the WT-like loop conformation (Figure 1). In the WT-like conformation the heme iron is accessible through a main access channel (Figure 1, green channel), proposed as the main entry route for hydrogen peroxide and located perpendicular to the heme plane [12]. The channel’s bottleneck is formed by Asp-143, Arg-232 together with Phe-248 and Leu-246 and blocks larger molecules such as glycerol from reaching the heme iron. Molecular dynamics simulations have shown that the D143A and D143A/R232A variants display a broader access channel and thus an increased solvent and substrate accessibility [13]. Fittingly only in these variants a glycine molecule is able to enter and occupy the sixth binding position of the Fe(III) ion at room temperature. Although glycine binding is also observed for WT and R232A *Kp*DyP at low temperature, this is most likely mediated by freezing-induced distortions of the heme cavity/entrance channels as already seen for other heme proteins [7,8,9,10]. Notably, this is not a common feature of all heme proteins, since ferric myoglobin is not binding glycine at room or low temperatures [36].

The heme architecture of WT, D143A and R232A *Kp*DyP was further analyzed at pH 7 in the presence of glycerol using the HYSCORE method. Glycerol was needed in this case as glassing agent to increase the electronic relaxation times and allow HYSCORE experiments. The overall HYSCORE features of the different *Kp*DyP variants are qualitatively in agreement with what was observed for other high-spin heme systems [37,38,39,40,41,42]. While the HYSCORE results are essentially the same for WT and D143A *Kp*DyP, subtle but significant difference were observed for R232A. Not surprisingly, a change was observed in the spectral region where contribution of remote ^14^N nuclei, such as the nitrogen of the distal Arg-232 are expected (Appendix A). However, the mutation of Arg-232 to Ala also influences the hyperfine interactions with the porphyrin ^14^N (Table 2). This does not seem to be correlated with the difference in *E*/*D* values, which are more alike for the two variants than for WT *Kp*DyP (Appendix A). X-ray crystallography has, however, shown that the heme cavity architecture is lost in the R232A variant, including displacement of a loop that holds Asp-143, thus altering the active site access channels [12]. The structural reorganization is also accompanied by the loss of a salt bridge involving a modified orientation of the heme propionate p7. It is thus interesting to note that the spin-density distribution in the heme plane is sensitive to these changes at the periphery of the heme moiety. So far, high-spin heme proteins are still scarcely studied by hyperfine spectroscopy allowing only limited comparison. Overall, the hyperfine values of the heme ^14^N nuclei of the *Kp*DyP are somewhat lower than those observed for other heme proteins (Table 2). All the investigated variants of *Kp*DyP show magnetically inequivalent heme nitrogens, in line with an earlier observation for metMb [39,56]. Quantum-chemical computations are needed to link the observed differences to the electronic structure of the ferric heme center, but state-of-the-art techniques, such as DFT, still fail to accurately reproduce the EPR parameters of these high-spin systems.

## 4. Materials and Methods

### 4.1. Protein Preparation

WT *Kp*DyP, and three variants (D143A, R232A and D143A/R232A) were recombinantly expressed and purified following the procedure described in [12]. Buffer exchange from a 50 mM phosphate buffer (pH 7.0) to a 50 mM borate buffer (pH 10.0) or a 50 mM glycine KOH buffer (pH 10.0) was done by dissolving the protein in a 10-fold excess of the new buffer solution and subsequent centrifugation in 0.5 mL Amicon centrifugation units (30 kDa cut-off). This was repeated three times, with an effective dilution of the initial buffer of 1 to 10,000 (UV-vis) or 1:400 (EPR) including the initial dilution to reach the desired protein concentration. 

### 4.2. Optical Absorption Spectroscopy

Absorption spectroscopy in the UV and visible region was performed using a Varian Cary 5E UV-Vis-NIR spectrometer combined with 10 mm quartz cells (Hellma Analytics, Kruibeke, Belgium). The spectra of all samples (protein concentration ≈ 20 µM) were recorded at room temperature for wavelengths ranging from 250 to 700 nm. In some cases, glycerol was added up to 25% of the total volume. All results were corrected with a baseline of the buffer (and glycerol) solution. 

### 4.3. EPR Spectroscopy

EPR was conducted on frozen protein solutions (enzyme concentration ≈ 500 µM) in the absence and presence of glycerol (25%). All samples were inserted in quartz EPR tubes (o.d = 4 mm for X-band and 0.6 mm for W-band) and flash frozen in liquid N_2_. 

X-band continuous-wave (CW) EPR experiments were performed on a Bruker ESP300E spectrometer (Bruker Biospin, Rheinstetten, Germany) operating at a microwave frequency of ca. 9.44 GHz equipped with a liquid-helium cryostat (Oxford Inc., Oxford, UK) to enable temperatures from 2.5 K up to room temperature. Calibration of the magnetic field was done using a Bruker ER035M NMR Gaussmeter. The EPR tubes were vacuum-pumped to 1 mbar prior to and during the experiments to remove excess of paramagnetic dioxygen. 

All spectra of the ferric proteins were recorded at 10 K under non-saturating conditions at 1 mW microwave power, 0.5 mT modulation amplitude and 100 kHz modulation frequency. 

X-band HYSCORE (hyperfine sublevel correlation spectroscopy) experiments [57] were carried out on a Bruker E580 Elexsys spectrometer (Bruker Biospin, Rheinstetten, Germany) (microwave frequency ≈ 9.74 GHz) equipped with an Oxford Instruments gas-flow cryogenic system to obtain an operating temperature of 4 K. The pulse sequence *π/2-τ- π/2-t_1_-π-t_2_- π/2-τ-echo* was performed using *t_π/2_* = 16 ns, *t_π_* = 32 ns and was repeated for four different τ-values (96, 104, 114 and 128 ns) at a magnetic field corresponding to *g_z_^eff^*. Matched HYSCORE [58] was performed at a magnetic field corresponding to *g_y_^eff^*, using *t_π/2_* = 16 ns, *t_π_* = 32 ns, *t_HTA_* = 24 ns and was repeated for two different τ-values (144 and 164 ns). *t_1_* and *t_2_* were varied in time steps of 16 ns, starting from 96 ns to 4896 ns. The HYSCORE spectra are baseline corrected using a third-order polynomial, apodized with a Hamming window and zero-filled. After Fourier transformation, the absolute value spectrum was calculated. HYSCORE measurements recorded with different τ-values were added together as indicated in the figure captions.W-band (93.98 GHz) electron spin echo (ESE)-detected EPR were performed on a Bruker Elexsys E680 spectrometer (Bruker Biospin, Rheinstetten, Germany) equipped with a standard single-mode cylindrical resonator from Bruker and a continuous-flow cryostat and superconducting magnet from Oxford Instruments. Using a *π/2-τ-π-τ-echo* sequence, the low-field part (<1800 mT) was obtained using π/2 (π) pulse lengths of 60 (120) ns and *τ* = 200 ns. The high-field part, representative for the low-spin signals, required π/2 (π) pulse lengths of 80 (160) ns and *τ* = 400 ns. 

Simulation of the experimental spectra was done using EasySpin [59], a toolbox developed for EPR simulations using MATLAB (MathWorks, Natick, MA, USA).

## 5. Conclusions

The spectroscopic study on WT *Kp*DyP and variants presented here draws attention to the impact of experimental conditions on the coordination state of heme proteins in general and more specifically of dye-decolorizing peroxidases. Buffer choice is often based on habit rather than reasoning. The case of the TRIS buffer exemplifies this; it is one of the most commonly used buffers, despite the fact that it strongly interacts with the peptide backbone [60]. Notably, other buffers such as HEPES have been shown to form radicals, rendering them unsuitable for studying redox active enzymes. Here, we have shown that a well-considered choice of buffer is important since buffer components are not only able to perturb the electronic architecture of the active site but can also be directly involved in binding. We have demonstrated the non-innocent role of buffer components by comparing WT *Kp*DyP and the D143A, R232A and D143A/R232A variants in widely used borate and glycine-KOH buffer at pH 10. The buffer molecule glycine is found to bind at high pH to the heme iron at the distal side of the heme when the bottle neck of the access channels is widened by an exchange of Asp-143 to Ala. When studying alkaline transitions in dye-decolorizing peroxidases (and other heme proteins), a careful choice of the buffer is thus needed. Indeed, hydroxo ligation at room temperature only occurs for the D143A variant at pH 10. Low-temperature EPR shows that the distal hydroxo ligand can be stabilized in this protein in three ways associated with different hydrogen bond strengths. Less frequently used buffers CHES and CAPS may provide good alternatives, but will require thorough investigation [61]. 

In line with earlier observations for other peroxidases, EPR reveals a large heterogeneity in the observed heme centers of WT *Kp*DyP and variants at different pH values. This heterogeneity is due to a combination of freezing effects, (undesired) ligation of exogenous molecules, pH and overall flexibility of the heme cavity. The freezing-induced artifacts can be prevented by a glassing agent, such as glycerol. Glassing agents are indispensable when performing pulsed EPR spectroscopy to reveal changes in the hyperfine values of the surrounding magnetic nuclei. However, our data show that glycerol may also influence the outcome of some of the experiments in a structure-dependent manner. Since glycerol can be accommodated in the access channel to the active site or in the heme pocket when the bottle neck radius of the channel allows so, it may prevent distal heme ligation, as observed for some of the variants at high pH. Conversely, as the effect of glycerol and buffer molecules is protein-variant specific, it may, when analyzed with care, be used to gain information on the accessibility of the heme region.

Both Asp-143 and Arg-232 are fully conserved in DyPs. The present work supports the role of this amino-acid pair in restricting the access to the heme cavity. Additionally, the observed strong ligation of glycine to all variants lacking Asp-143 suggests that is selectively inhibits substrate/ligand binding. With the exception of hydrogen peroxide no organic or inorganic substrates can enter the heme cavity nor approach the heme periphery for electron delivery. The ferric high-spin WT *Kp*DyP efficiently reacts with hydrogen peroxide, thereby forming a stable and unreactive Compound I with Asp-143 acting as proton acceptor in the heterolytic cleavage of hydrogen peroxide [12]. The role of Arg-232 in Compound I formation could not be evaluated so far because the distal heme cavity collapsed upon its exchange [12]. We previously proposed that Arg-232 supports Compound I formation by H-bonding of the hydroperoxyl anion (i.e., Compound 0 state) and heterolysis of the O–O bond electrostatically. That a hydroxo ligation is only possible in the presence of Arg-232, which presumably forms a hydrogen bond with the hydroxo-anion, strongly supports this hypothesis.

## Figures and Tables

**Figure 1 ijms-22-09849-f001:**
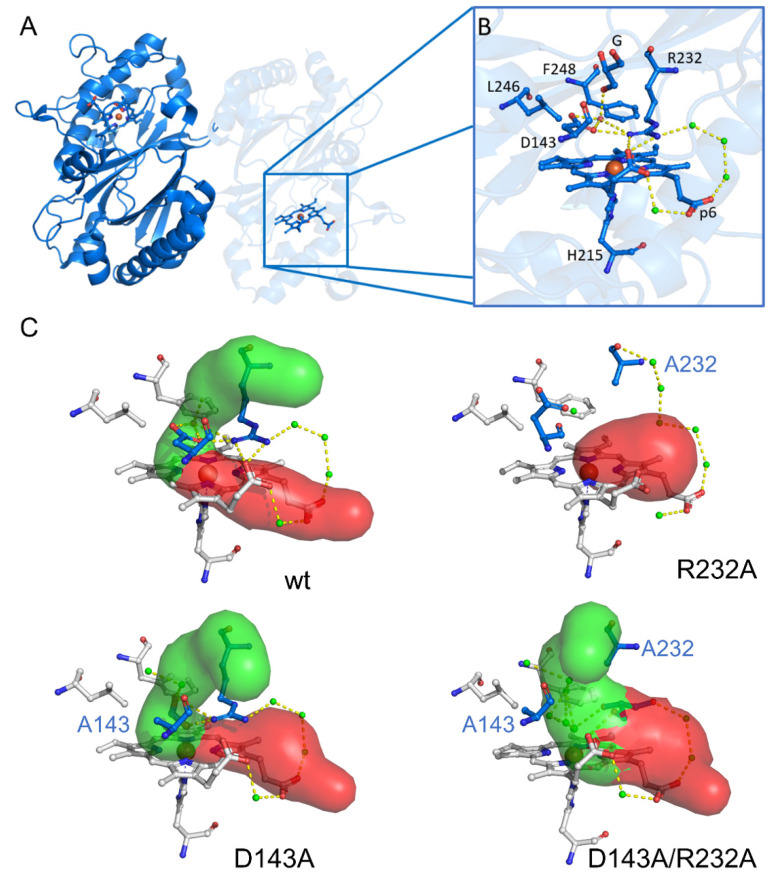
Overall crystal structure of the P-class dye-decolorizing peroxidase of wild-type *Klebsiella pneumoniae* (*Kp*DyP) and active site structures of WT *Kp*DyP (PDB entry 6FKS) and three variants: D143A (6FL2), R232A (6FKT) and D143A/R232A (6FIY). (**A**) Cartoon representation of the crystal structure of dimeric WT *Kp*DyP. (**B**) Active site structure of WT *Kp*DyP displaying key amino acids (H215, D143, R232, L246 and F248) of the heme cavity, as well as a glycerol molecule (designated G) and possible hydrogen bonds between water molecules (green). (**C**) Active site structures of WT *Kp*DyP and three variants, displaying the heme access channels, as determined by CAVER 3.0 [22].

**Figure 2 ijms-22-09849-f002:**
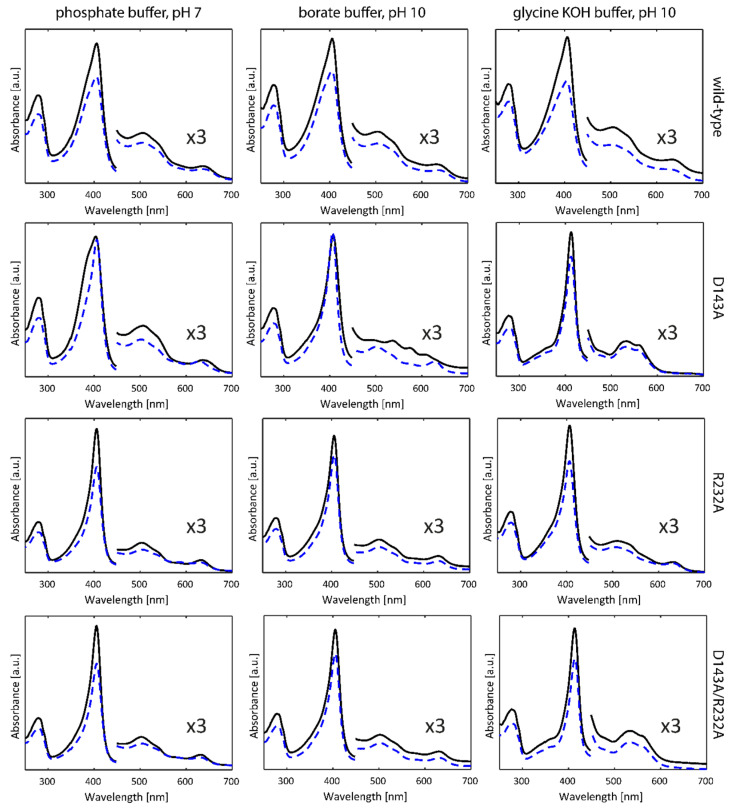
Optical absorption spectra of WT *Kp*DyP and variants in three different buffer systems without (black, full) and with glycerol (blue, dashed).

**Figure 3 ijms-22-09849-f003:**
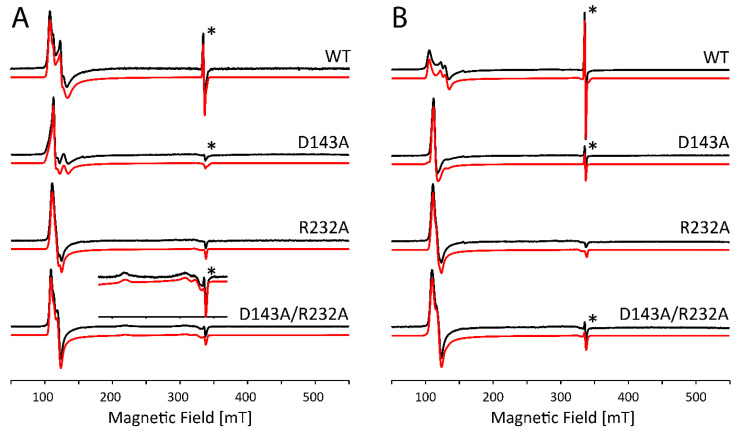
Experimental (black) and simulated (red) X-band CW EPR spectra of frozen solutions of WT *Kp*DyP and variants (c ≈ 500 µM) in phosphate buffer (pH 7.0) without (**A**) and with (**B**) glycerol. EPR simulation parameters can be found in Appendix A and the different contributions are displayed in Appendix A. An asterisk (*) indicates the position of a stable organic radical identified earlier in the resting-state of the protein [12]. The spectra are shown normalized to allow a facile comparison. An inset in (**A**) highlights the presence of a LS1 species in the double variant (5× amplified).

**Figure 4 ijms-22-09849-f004:**
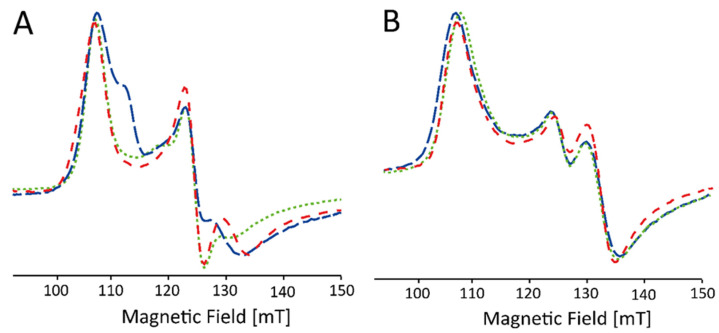
Low-field part of X-band CW EPR spectra of frozen solutions of WT KpDyP (c ≈ 500 µM) in phosphate buffer (pH 7.0). The different traces (indicated by different colors) represent different batches before (**A**) and after (**B**) the addition of glycerol. The spectra are shown normalized to allow facile comparison.

**Figure 5 ijms-22-09849-f005:**
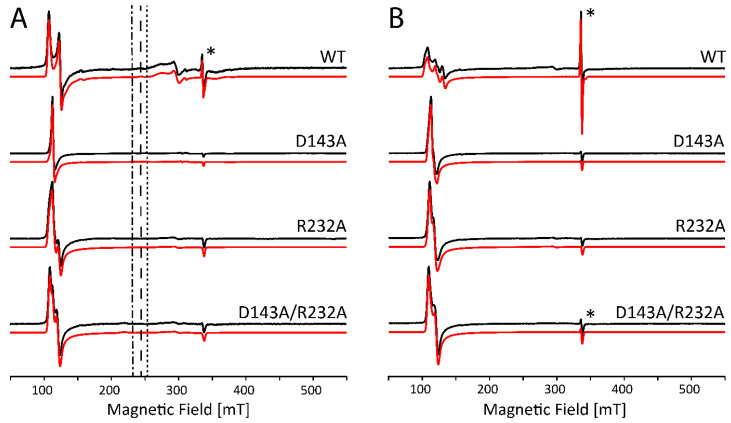
Experimental (black) and simulated (red) X-band CW EPR spectra of frozen solutions of WT *Kp*DyP and variants in borate buffer (pH 10.0) without (**A**) and with (**B**) glycerol. EPR simulation parameters can be found in Appendix A and the different contributions are displayed in Appendix A. An asterisk (*) indicates the position of a stable organic radical identified earlier in the resting-state of the protein [12]. The spectra are shown normalized to allow facile comparison. The position of the *g_z_*-component of the OH1′ (dashed-dotted), OH2/OH2′ (dashed) and the OH3′ (dotted) species is indicated, Appendix A shows an enlargement of the spectra in this magnetic field area to allow visual control of the presence of these species.

**Figure 6 ijms-22-09849-f006:**
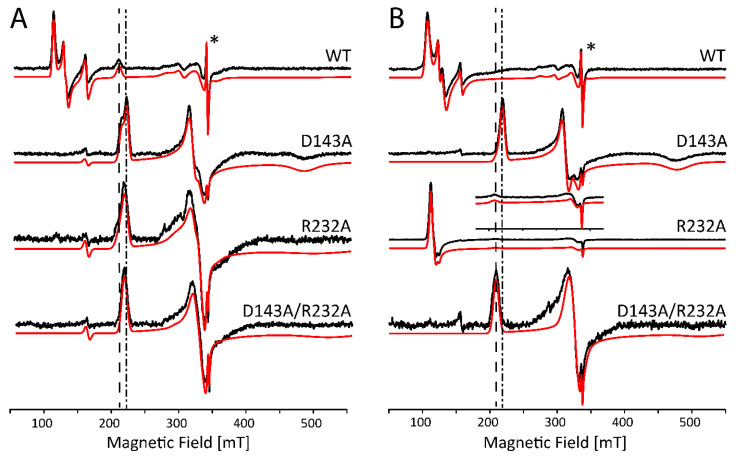
Experimental (black) and simulated (red) X-band CW EPR spectra of frozen solutions of WT *Kp*DyP and variants in glycine-KOH buffer (pH 10.0) without (**A**) and with (**B**) glycerol. EPR simulation parameters can be found in Appendix A and the different contributions are displayed in Appendix A. An asterisk (*) indicates the position of a stable organic radical identified earlier in the resting-state of the protein [12] and the *g_z_*-component of the Gly1 (dashed-dotted) and Gly2 (dashed) species is indicated as well. The spectra are shown normalized to allow facile comparison. An inset in (**B**) highlights the presence of a Gly2 species in the R232A variant (5× amplified).

**Figure 7 ijms-22-09849-f007:**
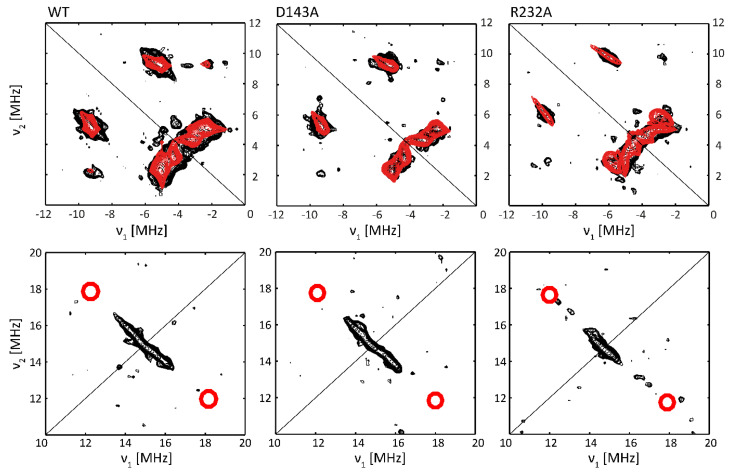
X-band HYSCORE of frozen solutions of WT, D143A and R232 *Kp*DyP variants in phosphate buffer (pH 7.0) and 25% glycerol, taken at 4 K and *B_0_* = 348 mT (magnetic field setting corresponding with *g_z_^ef^*^f^). The (-, +) quadrant (top panel) shows cross peaks stemming from the ^14^N heme nuclei and the (+, +) quadrant in the (10–20, 10–20) MHz region (lower panel) indicates interaction between the electron spin and nearby protons. The experimental spectrum (black) and the simulation (red) in the top panel are obtained by averaging over four *τ*-values (96 ns, 104 ns, 114 ns and 128 ns). Essential simulation parameters are highlighted in Table 2. The red circles (lower panel) indicate the position where the cross peaks due to distal water protons should appear in the case of an aquomet form [37].

**Table 1 ijms-22-09849-t001:** Experimental principal *g*-tensor values of the low spin species in frozen solutions from WT *Kp*DyP and the D143A variant in pH 10 borate buffer without glassing agent (experimental error ± 0.02 for *g_z_* and ± 0.05 for *g_x,y_*) and a representative selection of heme proteins with a distal hydroxo ligation at alkaline pH. HRP, horseradish peroxidase; C*c*P, cytochrome *c* peroxidase; swMb, sperm whale myoglobin; *Ce*GLB-33: GLB-33 from *C. elegans*; *Tf*Hb, hemoglobin from *Thermobifida fusca*.

	*g_z_*	*g_y_*	*g_x_*	Ref.
WT *Kp*DyP				
OH2	2.77	2.17	1.77	this work
D143A *Kp*DyP				
OH1′	2.89	2.12	1.78	this work
OH2′	2.728	2.15	1.772	this work
OH3′	2.67	2.214	1.82	this work
HRP	2.94	2.08	1.63	[30]
C*c*P	2.74	2.22	1.74	[31]
swMb	2.55	2.17	1.85	[29]
*Ce*GLB-33	2.62 2.845	2.20 2.12	1.815 1.69	[33]
*Tf*Hb	2.73 2.66 2.82	2.19 2.19 2.32	1.76 1.81 1.60	[32]

**Table 2 ijms-22-09849-t002:** Comparison between the z-component of the hyperfine (experimental error ± 0.03 MHz) and nuclear quadrupole values (experimental error ± 0.05 MHz) of the porphyrin ^14^N nuclei in WT, D143A and R232A KpDyP variants and those of other HS heme proteins. The z-direction is along the heme normal. n.r., not reported.

	|*A_z_^eff^*| [MHz]	|*Q_z_*| [MHz]
*Kp*DyP (this work) WT ^14^N_porf,1_	6.90	0.25
^14^N_porf,2_	6.70	0.28
D143A ^14^N_porf,1_	6.90	0.25
^14^N_porf,2_	6.70	0.28
R232A ^14^N_porf,1_	7.07	0.32
^14^N_porf,2_	6.95	0.28
metMb [39] ^14^N_porf,1_	7.42	0.33
^14^N_porf,2_	7.10	0.23
NGB [37] ^14^N_porf_	7.25	0.23
Dehaloperoxidase [38]		
^14^N_porf_	7.50	n.r.
CcmE chaperone [41] ^14^N_porf_	8.01	0.25

## Data Availability

The data presented in this study are openly available in Open Science Framework (OSF) at DOI 10.17605/OSF.IO/6B9WE.

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
