# Peer review of "In Vitro Heme Coordination of a Dye-Decolorizing Peroxidase—The Interplay of Key Amino Acids, pH, Buffer and Glycerol"

_ijms, 2021, doi:10.3390/ijms22189849_

Round 1
Reviewer 1 Report
General comment:
This manuscript, entitled “In Vitro heme coordination of a dye-decolorizing peroxidase the interplay of key amino acids, pH, buffer and glycerol,” authored by Nys et al., reports the interplay between structural flexibility, key amino acids, pH, temperature, buffer, and glycerol in spectroscopic studies of bacterial P-class DyPs concerning the poor peroxidase activity. Taking advantage of heme protein as an excellent probe for spectroscopic techniques, such as electron paramagnetic resonance and optical absorption spectroscopy, provide tools to study how the enzymatic function is linked to the heme-pocket architecture. This kind of approach is suitable for connecting biochemical and biophysical events to structure and function and is nice to answer fundamental biological questions in protein science. In my opinion, this is a valuable work and is suitable for publication in the International Journal of Molecular Sciences after the authors have addressed the following comments and major questions:
Major questions:
- To see the effects of pH, buffer, temperature, glycerol etc….why author choose the bacterial P-class DyPs peroxidase? The well-known standard Myoglobin, a heme protein, was not used for this kind of study. If the author is claiming this study will help understand protein structure affected by buffer condition in general.
- The insufficient peroxidase activity affected by the condition mentioned above? The structural changes may be minor alternatively not affect the function. Why didn’t the author show the peroxidase activity assay for all variants compared to the wild type in a different condition?
- How the tunnel analysis by CAVER support the mutation affects peroxidase activity or ligand entry? Comment - D->A mutation doesn’t show tunnel collapse (green), while R->A do – it seems Arg232 to Ala created a void in the pocket – which favor ligand entry but not the stabilization – correct?
- What is the reason for choosing Gly-KOH buffer? And how are you sure gly binds to the heme pocket? In my opinion -OH is binding to the heme pocket and stabilized by Arg232 strongly in the absence of Asp143.
- These experiments should be performed in a different buffer like just with KOH or NaOH buffer to avoid the conflict of gly or OH binding to the heme pocket. Compare to borate buffer, which devoid of OH may be the reason not to see the q-band splitting.
- Would you please summarize the effect of this component on wild type and mutants in a short paragraph?
Author Response
We would like to thank Reviewer 1 for the valuable comments that are now taken into account in the following way.
Major questions:
- To see the effects of pH, buffer, temperature, glycerol etc….why author choose the bacterial P-class DyPs peroxidase? The well-known standard Myoglobin, a heme protein, was not used for this kind of study. If the author is claiming this study will help understand protein structure affected by buffer condition in general.
ANSWER: The reviewer is correct that Myoglobin is a well-known standard for heme proteins. In fact, a detailed study of some of us on the buffer effects and on the effects of a variety of cryoprotectants on the alkaline transition of myoglobin detected by EPR is currently under revision and referenced in the manuscript (reference 36).
However, in order to more fully understand the effect of buffer conditions on protein structures and test the generic effects, it is necessary to study other, less frequently used systems. Here KpDyP, which is a novel fold and as a heme peroxidase has features, which are distinctly different from other heme proteins, represents an interesting target as it is structurally and biochemically well-characterized and may serve as an example for a large structural superfamily, which has gained significant interest in recent years. To more clearly state this we have amended the text as follows:
“KpDyP has been biochemically [12] and structurally [13, 14] well characterized and serves as a good model for this protein family, which has gained significant interest in recent years in regard to their biotechnological potential [15].” (line 56-58). Additional references ([13],[15]) were added in this section.
- The insufficient peroxidase activity affected by the condition mentioned above? The structural changes may be minor alternatively not affect the function. Why didn’t the author show the peroxidase activity assay for all variants compared to the wild type in a different condition?
ANSWER: As mentioned on page 1 and described in Pfanzagl et al. 2018 KpDyP has low peroxidase activity towards common peroxidase substrates and the active site variants are nearly inactive. The aim of this study was to investigate the effect of the introduced mutations on the structure of KpDyP and how a “wrong” choice of buffer and glycerol may affect this. The effect of the mutations on the peroxidase activity has been extensively studied and published previously (Pfanzagl et al. 2018).
- How the tunnel analysis by CAVER support the mutation affects peroxidase activity or ligand entry? Comment - D->A mutation doesn’t show tunnel collapse (green), while R->A do – it seems Arg232 to Ala created a void in the pocket – which favor ligand entry but not the stabilization – correct?
ANSWER: The reviewer is correct in that Arg232 seems to be essential for ligand binding, as the variants with the R232A mutation do not show any hydroxo ligation. However, glycine ligation seems to be possible in this variant, suggesting that here no stabilization of the ligand is required. Moreover, if only R232 is exchanged neither glycine nor hydroxo ligations is observable which together with the high K_D for cyanide (published in Pfanzagl et al. 2018) and the observed collapse of the main access channel suggest that in this case the accessibility is the limiting factor.
- What is the reason for choosing Gly-KOH buffer? And how are you sure gly binds to the heme pocket? In my opinion -OH is binding to the heme pocket and stabilized by Arg232 strongly in the absence of Asp143.
ANSWER: Glycine based buffers are very commonly used in studying heme proteins in the high alkaline region (above pH 9.5) as the choice of buffer systems for this pH regime is limited. Therefore, it is important to understand the effects of this buffer on heme centres.
The EPR and UV-vis spectra clearly show that the observed low-spin species is distinct from a hydroxo ligation. The UV-vis maxima of the glycine ligation are at 535 and 562 nm, whereas the maxima observed in case of hydroxo ligation are at 540 and 574 nm with a shoulder at 610 nm. We have amended the text to more clearly describe the differences:
“Both D134A and the double variant show maxima at 533 nm and 574 nm, which are distinct from the absorption maxima typical for OH- ligation observed for D134A in borate buffer (540, 574 and 610 nm).” (line 246-248)
Furthermore, the principal g values of the low-spin species observed in the D134A variant (Table S3, Gly1 and Gly2, Figure 6) are not compatible with those reported for hydroxo ligated ferric heme centres. More specifically, the maximum g values of the latter will be in the range of 2.6 to 2.89 (see Table 1), while the Gly1 and Gly2 are above 3.1 (Table S3). These g values are in line with the binding of a distal ligand via an amine group and glycine ligation.
- These experiments should be performed in a different buffer like just with KOH or NaOH buffer to avoid the conflict of gly or OH binding to the heme pocket. Compare to borate buffer, which devoid of OH may be the reason not to see the q-band splitting.
ANSWER: As mentioned before the choice of buffer systems is limited in the high alkaline region. However, we also observed OH-ligation when using phosphate buffer set to pH 9. KOH and NaOH alone have no buffer capacity and are unsuitable to study buffer effects.
- Would you please summarize the effect of this component on wild type and mutants in a short paragraph?
ANSWER: We have amended the section entitled alkaline transition to include a summary of the observed differences between the variants:
“In borate buffer, an alkaline transition is observed for D143A KpDyP, both at room temperature (Figure 2) and at 10 K (Figure 5A). Moreover, WT KpDyP exhibits a small fraction of hydroxo-ligated species in frozen solution (Figure 5A, Table S2). If R232 is missing no hydroxo-ligation can be observed at all, suggesting that R232 is required to stabilize this ligand. It is possible that in the case of WT KpDyP the hydroxo-ligated fraction is either too low at room temperature to be detected in the optical spectrum or that the OH- ligation observed at low temperature originates from packing forces on the heme induced by freezing as is the case for the alkaline state at neutral pH in HRP isozyme A2 [28,30].” (line 439-447)
Reviewer 2 Report
I liked the study by Nis et al describing some interesting effects of buffer choice and glycerol in the electronic properties of DyPs, which should be carefully assessed in other peroxidases. The methods and results are clearly presented, and the discussion is sound.
I do not have any heavy considerations in your paper, only a few questions about the design of the work.
- Why do you choose pH 7 and 10? Is there any reason regarding activity or stability for that? Basic pHs are interesting from an applied point of view, so I appreciate detailed insights into what happens at those pHs. Given the high concentration of protein needed for EPR I assumed pHs have to be carefully selected, but I would appreciate some discussion of what would happen at acidic pHs, relevant for other peroxidases.
- Why pH 7 and 10? 3 units of pH seems a lot in terms of redox potential
- Why do you choose those buffer molecules specifically? When dealing with metalloproteins buffer selection is of paramount importance due to metal chelation. Again, given the demanding amounts of protein for your work, a factorial experiment design (wide range of pHs with several molecules per pH) is out of the question, but do you think other alternatives for phosphate can have those non-innocent buffer effects? And other buffers classically used such as Tris?
- You briefly discuss that some effects on the activity have been observed, but I miss more detail into that if possible. Are pH 7 and 10 relevant for this type of DyPs or related DyPs?
Finally, a few minor comments on the manuscript.
- Figure 1 is essential to understand the work, but I find it difficult to read. Can you make the panels of the mutant variants bigger and clearer? Maybe setting up 2 top and 2 bottom, with the amino acids mutated in a different color than the non-mutated ones, will make the figure overall clearer.
- Line 465 it reads “bind occupy” it seems only one of those words is needed
- In the methods section, line 501, what is the excess of buffer used? How many times do you exchange buffer with amicon? It seems important based on the discussion that you completely rule out other buffer molecules in the mixture.
Thanks for the work
Author Response
We would like to thank Reviewer 2 for the valuable comments that are now taken into account in the following way.
MAJOR COMMENTS
- Why do you choose pH 7 and 10? Is there any reason regarding activity or stability for that? Basic pHs are interesting from an applied point of view, so I appreciate detailed insights into what happens at those pHs. Given the high concentration of protein needed for EPR I assumed pHs have to be carefully selected, but I would appreciate some discussion of what would happen at acidic pHs, relevant for other peroxidases.
- Why pH 7 and 10? 3 units of pH seems a lot in terms of redox potential
ANSWER 1 and 2: We chose to include pH 10 next to pH 7 for various reasons. In our earlier work (Pfanzagl, et al 2018), we had observed that D143A showed an alkaline transition at pH 10, that was not observed for the other variants.
As for the acidic pH range, KpDyP has previously been shown to be unstable below pH 5. Above pH 5 the UV-vis spectral characteristics are the same as pH 7, and all variants are in a HS state. Ligand binding due to deprotonation of the ligand molecules in the alkaline region renders this pH range more interesting also from a structural point of view. See also answer to question 4, for implemented additions to text.
- Why do you choose those buffer molecules specifically? When dealing with metalloproteins buffer selection is of paramount importance due to metal chelation. Again, given the demanding amounts of protein for your work, a factorial experiment design (wide range of pHs with several molecules per pH) is out of the question, but do you think other alternatives for phosphate can have those non-innocent buffer effects? And other buffers classically used such as Tris?
ANSWER: Indeed, the reviewer is correct, other buffer molecules may not interact with the heme active site. This said, this is exactly what we are highlighting with this study as we can clearly show that while some proteins may safely be studied in the tested buffers, already small alterations, such as the increase in accessibility of the active site may render a commonly used buffer unsuitable. Notably TRIS has been shown to strongly interact with the peptide backbone and thus represents one of the most unfavorable buffers. Others like HEPES or PIPES can form radical, which renders them unsuitable for studying redox active enzymes. A detailed investigation of CHES and CAPS, which would be good alternatives at the high alkaline region is underway but these buffers are not commonly used in enzyme assays. Here we wanted to focus on more widely used buffers. In addition, depending on accessibility also CHES and CAPS would be able to interact with the heme metal centre. This has been implemented in the text on p 19, as detailed in the answer to question 4.
- You briefly discuss that some effects on the activity have been observed, but I miss more detail into that if possible. Are pH 7 and 10 relevant for this type of DyPs or related DyPs?
ANSWER: While DyPs have been shown to be more active in the acidic region this is only true for a low pH range where protein stability is compromised and structural changes are difficult to interpret (below pH 4). Starting from pH 6 the pH dependent change in enzyme activity is minor. In addition, the tested variants are mutants of residues required for formation of Compound I and are highly inactive.
We have amended the text in several places to discuss the raised questions. We have included our reasoning for the choice of pH range on page 4 which now reads: “..in different buffers at neutral and high alkaline pH, as ligand binding is often enhanced in the alkaline region due to deprotonation. The buffers were chosen based on their widespread use and potential to influence the heme active site. We purposefully omitted the acidic pH range as no specific structural changes are expected other than pH induced unfolding starting from pH 5 [12].” (line 96-100)
Furthermore, inspired by the reviewer’s comment, we have elaborated further on our buffer choice and included notable references on the Goods buffer and TRIS, to point interested readers to alternative options.
Here the respective paragraph on page 19 now reads: “Buffer choice is often based on habit rather than reasoning. The case of the TRIS buffer exemplifies this; it is one of the most commonly used buffers, despite the fact that it strongly interacts with the peptide backbone [60]. Notably, other buffers such as HEPES have been shown to form radicals, rendering them unsuitable for studying redox active enzymes.” (lines 568-572)
and
“Less frequently used buffers CHES and CAPS may provide good alternatives, but will require thorough investigation [61]. “ (lines 584-585)
In these sections, two new references [60] and [61] were added.
MINOR COMMENTS
- Figure 1 is essential to understand the work, but I find it difficult to read. Can you make the panels of the mutant variants bigger and clearer? Maybe setting up 2 top and 2 bottom, with the amino acids mutated in a different color than the non-mutated ones, will make the figure overall clearer.
ANSWER: We have amended Figure 1 according to the reviewer’s suggestions.
- Line 465 it reads “bind occupy” it seems only one of those words is needed
ANSWER: We have amended the text accordingly. (now line 481)
- In the methods section, line 501, what is the excess of buffer used? How many times do you exchange buffer with amicon? It seems important based on the discussion that you completely rule out other buffer molecules in the mixture.
ANSWER: We have amended the text to include this information.
“...protein in a 10-fold excess of the new buffer solution and subsequent centrifugation in 0.5 ml Amicon centrifugation units (30 kDa cut-off). This was repeated three times, with an effective dilution of the initial buffer of 1 to 10000 (UV-vis) or 1:400 (EPR) including the initial dilution to reach the desired protein concentration.” (line 518-522)
Round 2
Reviewer 1 Report
This manuscript, entitled “In Vitro heme coordination of a dye-decolorizing peroxidase the interplay of key amino acids, pH, buffer and glycerol,” authored by Nys et al., reports the interplay between structural flexibility, key amino acids, pH, temperature, buffer and glycerol in spectroscopic studies of bacterial P-class DyPs concerning the poor peroxidase activity. Taking advantage of heme protein as an excellent probe for spectroscopic techniques, such as electron paramagnetic resonance and optical absorption spectroscopy, provide tools to study how the enzymatic function is linked to the heme-pocket architecture. This kind of approach is suitable for connecting biochemical and biophysical events to structure and function and is nice to answer fundamental biological questions in protein science. The authors made all those corrections and answered all the major questions asked by me. I am satisfied with the reply from the authors. In my opinion, this is a valuable work and is suitable for publication in the International Journal of Molecular Sciences.